# Design, Preparation and Properties of Polyurethane Dispersions via Prepolymer Method

**DOI:** 10.3390/molecules28020625

**Published:** 2023-01-07

**Authors:** Lijuan Sun, Hongmei Jiang

**Affiliations:** Shanghai Huafeng New Material R&D Technology Co., Ltd., Pudong New Area, Shanghai 200120, China

**Keywords:** emulsification, chain-extension, prepolymer method, emulsified water

## Abstract

A waterborne polyurethane dispersion for foamed synthetic leather base was designed and prepared using prepolymer method. There are many variables in the emulsification and chain-extension process of waterborne polyurethane (WPUR) dispersions prepared by prepolymer method. This work thoroughly evaluated the impacts of the steps of adding emulsified water, the temperature of the prepolymer and emulsified water, and concentration of ammonia water on WPUR dispersions by investigating the particle sizes/distributions and the mechanical stability. Changes in the temperature of the prepolymer and emulsified water, the concentration of ammonia water, and the step of adding emulsified water showed great impacts on the appearance and particle size of dispersions. Decreasing the temperature of the prepolymer and emulsified water and increasing the dilution ration of H_2_O to ethylenediamine (EDA) led to safe emulsification and dispersions with good appearance and narrow particle size distributions can be prepared. Surprising results were obtained by adding emulsified water in two steps, WPUR dispersions with a small particle size, narrow particle distribution and excellent tensile properties can be obtained. The optimized WPUR1 was applied to prepare water-based synthetic leather base after mechanical foaming, and the base presented the desired high performance, such as high folding resistance and peel strength.

## 1. Introduction

Owing to the stronger calls for environmental requirements and control of volatile organic compound (VOC) emissions of chemical industry, the development of aqueous polyurethane was promoted quite fast [1,2,3,4]. As for now, waterborne polyurethane has achieved multiple applications in areas of coating for various fibers and buildings [5,6], adhesives for alternative substrates [7,8,9], sealants for caulking materials [10,11], etc. Waterborne polyurethane is mainly produced through the acetone method. Over the past two decades, thousands of literatures and patents were issued to keep steady improvement of the production technology with excellent quality of waterborne polyurethane [12,13,14,15]. Compared with other methods that need lots of organic solvent such as acetone or 1-methyl-2-pyrrolidinone, the prepolymer method is truly solvent-free in the entire production process, showing more prominent advantages in the face of the existing green chemical policy [2,16,17,18].

The preparation of waterborne polyurethane usually undergoes a phase-inversion process, that is, emulsified water was added into polyurethane prepolymer under stirring, gradually completing the transition from water-in-oil to oil-in-water. In the phase-inversion stage, the system changes from polymer as a continuous phase to water as a continuous phase with changing of viscosity for the system. This often includes three stages. At the beginning, with the addition of water, the viscosity of the system will be reduced due to the disassociation of hydrophilic ions in the prepolymer chain by water molecules. With the further addition of water, the hydrophobic main chain of the prepolymer will be precipitated, and the viscosity of the system rises sharply. The viscosity of the system declines until the reverse transformation is completed [19]. In addition to the influence of molecular structure, the phase-inversion is also affected by the viscosity, temperature of the prepolymer and other parameters in the whole process. The phase inversion process will be directly reflected in the appearance, the particle sizes/distributions of the dispersions, and the final film performance. Organic solvent is barely used in the preparation of prepolymer through prepolymer method, resulting in that formation of stable polyurethane dispersion with high molecular weight would often be disturbed by high viscosity of the prepolymer during the dispersion step, especially in the case of low *R* value (molar ratio of NCO to OH) [20,21,22,23,24]. Considering viscosity control of the prepolymer, the temperature of prepolymer is usually higher than 45 °C in industrial production, and the *R* value of the prepolymer process is usually ≥2, which is much different from the acetone method, meaning there will be amount of active NCO groups in prepolymer. After dispersion step, it is the most important issue to control the extremely fast chain-extension reactions between NCO groups and NH_2_ groups [25,26]. It is well known that the reactivity of the NCO group toward NH_2_ is very high especially when the temperature is above 40 °C and the NCO group concentration is high, which has a high possibility of leading to emulsification failure. Molecular weight of polyurethane increases via the formation of urea linkages with NCO-terminated prepolymers and diamines in the chain extension step [25,27]. Therefore, for the prepolymer method, prepolymerization process is easy to control and not a difficulty, the emulsification and amine chain extension stage is difficult to control [28] because of the above reasons, which directly determines whether the emulsion can be obtained or not.

Existing literatures for the prepolymer method mainly focus on the relationship between the structure and properties of the prepolymer or emulsification stage, including the effect of NCO/OH, DMPA content, hard segment content, mixed diisocyanate, and the neutralization degree, etc. [18,27,28], while a small amount of literature has studied the aspect of equipment [4]. However, detailed studies regarding the effect of operating variables of emulsification and chain-extension process for prepolymer method are rarely reported in the literature. For promoting the industrialization of waterborne polyurethane by prepolymer method, it is necessary to investigate the regularity of emulsification and amine chain extension process.

To investigate the influence of variables in emulsification and amine chain extension processes, we designed a waterborne polyurethane via prepolymer method. The effects of the temperature of the prepolymer and emulsified water, the concentration of ammonia water, and the steps of adding emulsified water were investigated and characterized with particle size and distribution, consequently dispersion appearance and properties were discussed. The application of this obtained waterborne polyurethane finally used in preparing a water-based synthetic leather base through a mechanical foaming process, the foam appearance and application properties, including folding resistance and peel strength, were confirmed to meet the requirements for possible future applications. 

## 2. Results and Discussion

### 2.1. Characterization of Prepolymer

The synthesis of WPURs is presented in Figure 1. For the purpose of fully studying the prepolymer process and obtaining WPUR applied in foam layer of synthetic leather, polypropylene glycol (PPG) with a conventional molecular weight (Mn = 2000) was used as the soft segment to take advantage of its good molecular flexibility and outstanding low-temperature performance, while aliphatic isocyanate (IPDI) was served as the main part of a hard segment. Carboxylic acid type hydrophilic chain extender DMPA was introduced into the prepolymer as a hydrophilic chain extender, another small molecular chain extender (MPO) was to consume isocyanate to achieve the *R* value of prepolymer falls in the range of 2 to 3. Since there is no acetone to dilute the prepolymer in the whole process, the *R* value of the prepolymer is greater than 2 to ensure the appropriate molecular weight to ensure that the viscosity of the prepolymer is suitable for subsequent emulsification. 

First, the prepolymer was synthesized by a one-pot process through incubating dehydrated polypropylene glycol (PPG 2000), diisocyanate (IPDI), MPO and DMPA with DBTDL as catalyst at 90 °C for 3 h. Furthermore, after the carboxylic acid group in the prepolymer is neutralized to achieve ionization situation, the ionized carboxylic acid group tends to be hydrophilic. Here, the *R* value of the prepolymer is 2.644, and the viscosity of the prepolymer at different temperatures was shown Figure 1a, which are all within the viscosity range (<5000 mPa·s) that is easy to undergo phase inversion during emulsification and dispersion. The next step is to emulsify the prepolymer to prepare the polyurethane dispersion with the synergistic emulsification of SDBS. The FTIR spectra of polyurethane prepolymer and WPUR1 were shown in Figure 1b. The absorption peaks at 2268 cm^−1^ in PU prepolymer is very strong, corresponding to the stretching vibration of NCO group. In the FTIR curve of WPUR1, the NCO peak disappeared, presenting all of NCO groups had reacted after emulsification and chain-extension processes. The absorption peaks at 3365 and 1530 cm^−1^ were attributed to stretching and bending vibrations of N-H groups in urethane group. The absorption peaks at 1720 cm^−1^ were corresponding to the C=O groups in urethane groups of WPUR1.

The following mainly explores the influence of different dispersion conditions, including the addition of emulsification water, the temperature of the prepolymer, the temperature of the emulsification water, and the dilution concentration of the chain-extended amine solution.

### 2.2. Effect of Water-Adding

According to literature reports, there are two ways of adding emulsified water when dispersing polyurethane prepolymer for the acetone method. One is that the prepolymer is rapidly added with water under mechanical stirring to complete the phase inversion to obtain an oil-in-water polyurethane dispersion. Another is the prepolymer added to the emulsified water while stirring to complete the phase inversion of the mixture. In industry, waterborne polyurethane is often produced by adding all the emulsified water all at once to the prepolymer under stirring [29,30,31]. 

For the prepolymer method, since there is a large amount of NCO groups in the prepolymer, after the subsequent phase inversion, a large amount of amine solution will be used for chain-extension, and the amine reacts violently with NCO during chain extension, often leading to a rapid increase in the viscosity of the dispersion. Thereby the progress of the chain extension of the amine affects the appearance of the dispersion. From the perspective of decreasing the viscosity after amine chain extension reaction, the two-step adding emulsification water method was explored in this study, the particle sizes/distributions and the mechanical stability after centrifugation of emulsions were used to analyze the effect of the two methods of adding water, and the results were shown in Appendix A. 

The effect of water-adding on emulsion quality was studied first. From Figure 2a, comparing to WPUR2 and WPUR4, WPUR1 and WPUR3 provided brighter blue light and better appearance. The particle size of these WPUs were shown in Figure 2b–e. When the prepolymers were directly dispersed through adding water all at once, WPURs were attained with large particle sizes of 294.9 and 278.4 nm and broad distribution of 0.565 and 0.470 for WPUR2 and WPUR4, respectively. However, when the emulsified water is added in two parts, most of water was added before the phase inversion to obtain an oil-in-water dispersion, and a small part was added at the amine chain extension process WPURs with appearance of blue light (Figure 2a) and smaller particle size of 170.6 and 164.8 nm and narrower distribution of 0.157 and 0.114 for WPUR1 and WPUR3, respectively, were finally obtained. In addition, from the results of CPH (Appendix A), the WPURs obtained by adding water in two steps were also more stable.

From the above results, it can be speculated that in the process of viscous amine chain-extension, adding some water can effectively reduce the intensity of the reaction, and a relatively slow chain extension reaction can help to obtain WPURs with good appearance. Subsequent studies were performed by two-step addition of water to prepare WPURs.

### 2.3. Effect of the Temperature of Prepolymer and Emulsification Water

As mentioned above, for the prepolymer method, there is a large amount of active NCO groups left in the prepolymer before emulsification. It is crucial to control the reaction of NCO with water and amine in the emulsification and following chain extension processes. Due to the huge reaction speed difference, the consumption of NCO by water is not conducive to the diamine chain extender. At the same time, the amine chain extension reaction is violent enough and will bring the risk of explosion, then WPU cannot be obtained. Therefore, it is of great exploratory significance to intervene in the unintended reaction of NCO under temperature control.

Here, the effects of different prepolymer temperatures and emulsification water temperatures on the emulsification and chain extension processes were investigated, similarly, the particle size and distribution and the mechanical stability after centrifugation of emulsions were used to analyze the effect of the temperature of prepolymer and emulsification water. Since amine chain extension follows phase inversion, here the phenomena of both processes were discussed together. From Appendix A, it can be seen that when the emulsification water temperature was 5 °C and the prepolymer temperature was within 45–60 °C, dispersions (WPUR1, WPUR3, WPUR5, WPUR6, WPUR9 and WPUR10) can be obtained. Figure 3 gives the corresponding particle sizes and distributions of these dispersions. It can be seen WPU1 still has the best particle size and distribution comparing with WPUR3, WPUR5, WPUR6, WPUR9 and WPUR10. When the prepolymer temperature reached 65 °C and above, the amine chain extension cannot be completed normally, and even unable to obtain dispersions with normal appearance. The case where the prepolymer temperature was lower than 45 °C was not further studied here because the viscosity of the prepolymer at a lower temperature was no longer suitable for phase inversion. The temperatures during the phase inversion and amine chain extension process were observed and recorded. When amine and water were added to the dispersion system, the viscosity of the dispersion rapidly rose and the temperature of the whole system increased by about 8–13 °C for the existing volume. However, for the experiments with a lower prepolymer temperature, when the low temperature emulsified water at 5 °C was added to the prepolymer, due to the low temperature of the whole system, the temperature-increase brought by the amine chain extension did not result in detonation problem. Likewise, the effect of emulsification water temperature was similar. From the experimental results, when the temperature of the prepolymer was kept at 45 °C, the dispersion cannot be obtained when the temperature of the emulsified water was greater than or equal to 20 °C. While the temperature of the prepolymer was lower than 20 °C, the dispersion can be obtained successfully. In fact, the phenomenon observed in the process was also similar. It is the risk of explosion caused by the high temperature of the system during chain extension of the amine. At the same time, considering that the reaction activity of NCO and H_2_O increases when the system temperature was high, the degree of amine chain extension reaction will not proceed as expected due to the competition of water. 

From the above results, it can be concluded that both the temperatures of prepolymer and emulsified water has an important influence on phase inversion, and finally affect the appearance of lotion. The comprehensive selection of prepolymer temperature was set at 45 °C, and emulsified water temperature of was set at 5 °C, it is a suitable parameter for proper chain extension reaction and good emulsions. 

### 2.4. Effect of the Concentration of the Diamine Chain-Extender 

The concentration of the diamine aqueous solution also has an important influence on the progress of the amine chain extension. Generally, for the acetone method, the diamine is usually diluted with a small amount of emulsified water to 1:3 to 1:2 (mass ratio) during the production process to prevent partial high amine concentration in the system which would cause adverse effects, including the inability of the chain extension reaction to proceed uniformly and the high local reactivity, ultimately affect the performance of WPURs. Here, the dilution ratio of the chain-extension amine to water, including, 1:5 (WPUR1), 1:4 (WPUR13), 1:3 (WPUR14), 1:2 (WPUR15) and 1:1 (WPUR16), were investigated (Appendix A). From Figure 4a–e, for the dispersions with EDA:H_2_O of 1:5 (WPUR1), 1:4 (WPUR13), 1:3 (WPUR14), the particle size-distributions were very narrow. The tensile properties of films prepared using these WPURs were also examined at the same time (Figure 4f). Overall, the strength of these films showed a downward trend with the increase of amine dilution, while WPUR1 showed the best elongation. Appendix A shows the TG (relationship between mass change and temperature) curves of different WPUR films. The thermal decomposition processes of these polyurethane films are similar; however, WPUR1 still exhibits the best thermal decomposition resistance. According to the above results, the dilution ratio of amine affects the chain extension process of amine, and further affects the particle sizes and film performance of WPURs. In this study, the optimal dilution ratio of amine and water was 1:5.

### 2.5. Application Performance of Waterborne Synthetic Leather Base

Waterborne synthetic leather base is the precursor of the preparation of waterborne leather products. Synthetic leather would be obtained by further scraping a layer on the surface of base, so the base will ultimately play a decisive role on the application properties of leather products. In the field of synthetic leather, the fineness of foam, the peel strength and folding resistance of base are mainly concerned, because fine and uniform foam will bring a comfortable and soft touch. Folding resistance is an effective indication of the dynamic fatigue life of synthetic leather, it is the times of the synthetic leather when it was bent to breakage or cracks appeared. For shoe leather, the number of folding-resistance usually means the maximum number of bends when the shoe is worn, that is, the service times of the shoes. Good folding resistance indicates creases will not be left on synthetic leather after bending when used, while the bonding fastness of the WPU layer and substrate is characterized with peel strength. The peel strength also indicates the service life of synthetic leather to a certain extent, for example, compared with high peel strength, synthetic leather with low peel strength will have the delamination of polyurethane layer and substrate earlier when used, thereby shortening the service life of synthetic leather. According to the Industry standard of light Industry of the People’s Republic of China (QB/T 1646-2007), synthetic leather meets the optimal index of III when the folding resistance of synthetic leather is greater than 200,000 times and the peeling strength is above 49 N.

After the above optimizations, WPUR1 was confirmed as optimal and further used to prepare waterborne polyurethane synthetic leather base. SEM images of foaming base prepared with WPUR1 were given in Figure 5. From Figure 5a, some foam holes with similar pore sizes distributed on the surface of the base without obvious irregularities. The side section image (Figure 5b) of the base more clearly presented the morphology of foams, these closed foams were evenly distributed, the connection between these foam walls was tight, and there was no obvious cell fusion, that is, the collapse of the foam wall. This indicated that these foams firmly supported the base, which will have a comfortable and soft feel. The base was further bent until cracks appeared, and eventually the base broke when it was bent approximately to 230,000 times, while and the peel strength tested here is 58 N (Figure 5c) indicating the force used to tear the foam wall of the base. The above results show that the prepared base has excellent fundament application performances.

## 3. Materials and Methods

### 3.1. Materials

Polypropylene glycol (PPG, Mn = 2000) was supplied by Dow Co. (Shanghai, China). Isophorone diisocyanate (IPDI) was supplied by Evonik Co. (Shanghai, China). Dimethylol propionic acid (DMPA), dibutyltin dilaurate (DBTDL), 2-methyl-1,3-propanediol (MPO), triethyl amine (TEA) and ethylene diamine (EDA) were received from Sigma-Aldrich Co. (Shanghia, China). Sodium dodecyl benzene sulfonate (SDBS, 50%wt) was purchased from Sigma-Aldrich Co. Thickener UH-450VF was purchased from Adico (Shanghia, China). A water-based foaming agent EP112 was purchased from Evonik (Shanghia, China). PPG was heated to remove water in the vacuum drying oven at 120 °C for 1 h before use. Other materials were used as received. Deionized water was used for all emulsion experiments.

### 3.2. Preparation of NCO-Terminated Polyurethane Prepolymer

Preparation of prepolymer was carried out in a three-neck round-bottom flask equipped with mechanical stirrer, reflux condenser, thermometer and nitrogen gas inlet. PPG 2000 (200 g, 100 mmol) was placed in the flask and was dried in a vacuum at 120 °C for 1 h. After the temperature was decreased to 70 °C, IPDI (80 g, 360.36 mmol) were added slowly into the flask, and then the mixture was heated up to 90 °C with continuous stirring. The reaction was performed for 1 h. Then, MPO (3.85 g, 28.73 mmol), DMPA (5 g, 37.31 mmol), DBTDL (0.2 g) were added, and the reaction was kept for another 3 h at 85 °C to obtain an NCO-terminated polyurethane prepolymer until the realization of the theoretical NCO content using a standard di-*n*-buthylamine back-titration method (ASTM D2572-97). 

### 3.3. Dispersion and Amine-Chain Extension 

Dispersion of polyurethane was accomplished by adding deionized water to neutralized prepolymer. Before dispersion, add 3.77 g (36.38 mmol) of TEA to the above NCO-terminated prepolymer with stirring for 5 min to neutralize the prepolymer. Dispersion of polyurethane was accomplished by adding the mixture of water and surfactant (SDBS, 1.0 wt.% based on prepolymer) to the neutralized prepolymer at agitation rate of 2000 rpm. After 5 min, EDA solution (10.39 g, 17.31 mmol) was added over a period of 3 min, and amine-chain extension was carried out for another 10 min to obtain a dispersion with a solid content of 40%. 

For the one-step water addition method, deionized water was added all at once in the dispersion stage to obtain a dispersion with solid content of 40%; for the two-step water addition method, most of the deionized water was pre-added in the dispersion stage to obtain a 50% solid dispersion firstly, after EDA solution was added, a small additional portion of deionized water was added again to obtain a final dispersion with 40% solid content.

By changing the water-addition steps (one-step or two step), temperature of prepolymer (45 °C, 50 °C, 55 °C, 60 °C, 70 °C) before dispersion, temperature of deionized water (5 °C, 10 °C, 15 °C, 20 °C, 25 °C) for dispersion, dilution ratio of ammonia and deionized water (1:1, 1:2, 1:3, 1:4, 1:5), a series of WPUR dispersions (WPURs) were obtained. The sample designations of WPURs are shown in Appendix A. The temperature of the prepolymer was controlled by cooling the prepolymer from 85 °C, and the temperature of emulsified water was controlled by adding low-temperature deionized water and ice.

### 3.4. Preparation of WPUR Films

A total of 200 g of WPURs were weighed and stirred under 300 rpm; then, a small amount of UH-450VF was added for several times until the viscosity achieved 10,000 mPa·s. After that, smear the above slurry on mirror release-paper using a 600 μm spatula, and then the release-paper was transferred to a constant temperature oven at 50 °C for drying. After drying, the WPUR films were obtained for examining the tensile properties.

### 3.5. Preparation of Water-Based Synthetic Leather Base

Firstly, 200 g of WPUR1 and 10 g of EP112 were weighed, and the mixture was stirred at 300 rpm for 5 min. Then, a small amount of UH-450VF was added several times until the viscosity achieved 15,000 mPa·s, and then increasing the rotating speed to 1500 rpm for foaming to obtain the volume was 1.5 times of the original volume. Finally, the above slurry was smeared on the base cloth using a 1000 μm spatula, and then the base was transferred to an oven for drying. The drying procedure was to first keep the oven at 80 °C for 3 min, and then heat the oven to 140 °C and kept it for another 7 min. After drying, the water-based synthetic leather base was prepared.

### 3.6. Measurements

Fourier transform infrared spectroscopy (FTIR) experiments were carried out on a Nicolet iS10 FTIR spectrometer (Nicolet, Waltham, MA, USA). Particle sizes were determined using a Malvern Panalytical Zetasizer Nano ZSE light-scattering ultrafine particle analyzer (UK). The sample was diluted to the required concentration with distilled water before measurement. Viscosity measurements of the dispersions were performed using NDJ-9S viscometer (Shanghai, China), at a shear rate of 100/s at 25 °C. Tensile properties and peel strength tests were performed with Instron 3367 (Shanghai, China). Dispersion stability of WPURs was investigated by the centrifugal sedimentation method after the dispersion (10 mL) was centrifuged at 3000 rpm for 20 min on a KA 1000 (Shanghai, China), here the height of centrifugal precipitate (CPH) was used to evaluate the emulsion stability of WPURs. The fold-resistance test for the base was performed on GT-7071-BN (Wuhan, China). Scanning electron microscopy (SEM, Shanghai, China) images of the surface and side section for synthetic leather base were obtained with a SIGMA300.

## 4. Conclusions

The current study highlights the effect of water-adding method, the temperature of prepolymer and emulsification water and concentration of the chain-extended amine during emulsification and chain-extension process of a given formulation prepared by prepolymer method. The dispersion was mainly characterized by appearance, particle size and particle size distribution. By adding the emulsified water twice, WPURs with excellent appearance can be more easily and safely obtained. Both decreasing the temperature of the prepolymer and emulsification; water increased the safety of amine chain extension process. The concentration of the chain-extended amine was found to be critical for the properties of WPURs. WPUR1 prepared by optimized emulsification and amine chain extension conditions exhibits good mechanical properties and was further used to prepare water-based synthetic leather base after mechanical foaming, and the base was found to present the desired high performance.

## Data Availability

The data presented in this study are available in this article.

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
