# Peer review of "Design, Preparation and Properties of Polyurethane Dispersions via Prepolymer Method"

_molecules, 2023, doi:10.3390/molecules28020625_

Round 1

Reviewer 1 Report

The paper entitled “Design, preparation and properties of polyurethane dispersions via prepolymer method” focus on waterborne polyurethane dispersion for foamed synthetic leather. They are good candidates for coatings due to its very low organic content, easily handling property, and wide range of film hardness. The article shows and proves the chemical structure in detail.

1. As a recommendation to the authors, figure 4f should be a separate figure.

2. There is no description of the SEM method in the experimental part.

3. For convenience, readers need to accurately show the scientific novelty of the work.

4. In the introduction part, it is necessary to expand the literature review. It is necessary to make a comparison with numerical values.

As a result, I will recommend the publication of this manuscript after major revision. 

Reviewer 2 Report

in an attached file

Author Response

Please see the attechment.

Round 2

Reviewer 1 Report

The authors have answered satisfactorily the reviewer's comments, the paper  can be published.